# In Vitro Studies of Graphene for Management of Dental Caries and Periodontal Disease: A Concise Review

**DOI:** 10.3390/pharmaceutics14101997

**Published:** 2022-09-21

**Authors:** Mohammed Zahedul Islam Nizami, Iris Xiaoxue Yin, Christie Ying Kei Lung, John Yun Niu, May Lei Mei, Chun Hung Chu

**Affiliations:** 1Faculty of Dentistry, University of Hong Kong, Hong Kong SAR 999077, China; 2Faculty of Dentistry, University of Otago, Dunedin 9054, New Zealand

**Keywords:** caries, remineralization, graphene, periodontal, regeneration

## Abstract

Graphene is a single-layer two-dimensional carbon-based nanomaterial. It presents as a thin and strong material that has attracted many researchers’ attention. This study provides a concise review of the potential application of graphene materials in caries and periodontal disease management. Pristine or functionalized graphene and its derivatives exhibit favorable physicochemical, mechanical, and morphological properties applicable to biomedical applications. They can be activated and functionalized with metal and metal nanoparticles, polymers, and other small molecules to exhibit multi-differentiation activities, antimicrobial activities, and biocompatibility. They were investigated in preventive dentistry and regenerative dentistry. Graphene materials such as graphene oxide inhibit cariogenic microbes such as *Streptococcus mutans*. They also inhibit periodontal pathogens that are responsible for periodontitis and root canal infection. Graphene-fluorine promotes enamel and dentin mineralization. These materials were also broadly studied in regenerative dental research, such as dental hard and soft tissue regeneration, as well as periodontal tissue and bone regeneration. Graphene oxide-based materials, such as graphene oxide-fibroin, were reported as promising in tissue engineering for their biocompatibility, bioactivity, and ability to enhance cell proliferation properties in periodontal ligament stem cells. Laboratory research showed that graphene can be used exclusively or by incorporating it into existing dental materials. The success of laboratory studies can translate the application of graphene into clinical use.

## 1. Introduction

Graphene is a two-dimensional mono-atomic sp2 hybridized carbon-based nanomaterial known as the thinnest and strongest element in existence [1]. Of all the generic compounds and nanomaterials used in antimicrobial and regenerative research, graphene and its derivatives have attracted the attention of researchers in recent decades. It has a high surface area, excellent electrical and thermal conductivity, mechanical properties, low coefficient of thermal diffusion, and a significantly high aspect ratio. These features make it outstanding in a number of potential applications in a variety of fields, from engineering to biology [2,3,4,5,6,7,8]. Graphene and its derivatives can act as good substrates for diffusion, dispersion, and stability of many antimicrobial nanoparticles (i.e., copper, silver, iron, magnesium, calcium, titanium dioxide, zinc oxide, etc.) [9,10,11,12,13,14,15]. Moreover, graphene and its derivatives are suitable candidates for biological/chemical functionalization [16,17]. Their biocompatibility received great attention in the research on their potential applications in the biological, biomedical, medical, and dental fields [18]. Dentistry has a broader aspect in preventing and restoring decayed or lost teeth and dental tissues. Graphene’s potential antibacterial and tissue regenerative properties were widely used in various dental research fields [19,20,21]. Graphene is especially used in caries and periodontal disease management using its antibacterial properties, dental hard and soft tissue remineralization capacities, regeneration abilities, as well as its periodontal tissue and bone regeneration properties [20].

Dental caries is the chronic local damage of dental hard tissue (enamel, dentin, and cementum) that acidic byproducts of bacterial metabolism of dietary carbohydrates often cause [22], and periodontal disease is the inflammation of periodontium (gums, periodontal ligaments, and alveolar bone surrounding the teeth) [23]; both are associated with microbes. Generally, there is a balance between microfloral and microbial colonization and the oral microenvironment [24]. Unpleasant consequences occur when this balance is disturbed. The main cariogenic microbe is *Streptococcus mutans*, which generates organic acids, thus reducing the oral pH level and leading to demineralization of the dental hard tissue surface [25]. On the other hand, *Porphyromonas gingivalis* and *Fusobacterium nucleatum* are responsible for gingivitis and periodontitis, respectively [26]. Several strategies have been investigated, established, and employed in different communities and have brought beneficial implications for many world populations to manage these diseases. Preventive measures against dental caries and periodontal diseases have been remarkably improved over the last few decades with the advancement of nanotechnologies and nanomaterials. Whatever the immense struggles, a large number of the population still suffer from these diseases and eventually lose teeth [27].

Generally, humans cannot reproduce or regenerate or regrow teeth or tooth tissues. Although oral and soft tissue can be repaired, the regeneration or repair of hard tissues (enamel, dentin, and bones) is inadequate or sometimes impossible. Interestingly, in addition to biomaterials sciences, preventive and regenerative dentistry is also advancing well. Preventive and regenerative dentistry research mostly focuses on preventing dental caries and periodontal disease. Simultaneously, they are focused on restoring lost tooth tissue because of caries and or periodontal diseases. These days, the high incidence of periodontal diseases is a major concern [28]. Researchers are exploring real-time solutions for lost tooth tissue and bone, but it is a great challenge to achieve the outcome [29,30,31]. Regenerative dentistry and tissue engineering are now the most challenging research topics in this field. With these research advancements, it is not only periodontal diseases and tooth loss but also surgical resection of the maxillofacial hard and soft tissue (jawbone, tongue), due to trauma or oral cancers, that will also benefit.

Regardless of the challenges ahead, the latest advances in nanotechnology have played a biomimetic role and have shown tremendous potential in dental hard and soft tissue regeneration. Various nanomaterials are being added continuously and have produced many clinical benefits in dentistry using tissue engineering properties, which include: the advanced treatments of caries and periodontal diseases, bone regeneration, feasible biological tooth repair after caries, and is probably advancing towards regrowing entire lost teeth [32,33]. At this stage, graphene materials have shown potential in several in vitro studies that can be translated to in vivo and clinical settings. Therefore, the perspective of graphene and its derivatives in caries and periodontal disease management are concisely discussed in this review.

## 2. Materials and Methods

### 2.1. Graphene Derivatives Used for Caries and Periodontal Disease Management

Graphene is an excellent nanomaterial; however, in some cases, graphene may not adhere to the specific properties required for certain applications. Therefore, the functionalization of graphene comes to light in current research with improved properties known as graphene derivatives [34]. Studies demonstrate that graphene can be easily functionalized (chemically) using its functional groups [35]. Various graphene derivatives have been reported in recent studies. Namely, *graphene oxide, reduced graphene oxide, graphene, graphone, flurographene, graphyne, graphdiyne, doped graphene,* and *graphene quantum dots* were well reported [36,37,38]. Among them, graphene, graphene oxide, reduced graphene oxide, and graphene quantum dots were mostly studied in the biomedical research field [39]. In our literature search, we noted that these derivatives were also well investigated in caries and periodontal research.

Graphene oxide is the oxidized graphene-based sheets with several oxygen-functional groups (i.e., hydroxyl epoxy and carboxyl groups), which provide the covalent or noncovalent combination of graphene oxide with nanoparticles and nano-biomolecules. Therefore, graphene oxide properties can be tuned by changing its molecular structure by functionalization of its oxygen functional group [40,41]. Moreover, graphene oxide still can maintain the thinnest atomic structure compared to graphene, thus extending more active sites for functionalization [42].

Graphene oxide is a promising carrier for biomolecules and drugs. It can enhance biomaterials’ bioactivity, sustainability, and mechanical performance [20]. At the same time, reduced graphene oxide is another interesting derivative of graphene. It can be synthesized by the removal of oxygen functional groups of graphene oxide through electrochemical, thermal, photochemical, or microwave reduction methods [43,44]. However, oxygen-functional groups can be found on the reduced graphene oxide surface with a predominance of the carboxyl group and other defects [45,46]. On the other hand, graphene quantum dots is another graphene derivative consisting of mono- or few-layer zero-dimensional graphene, exhibiting dynamic properties applied to biomedical research [47,48,49].

Interestingly, all these graphene derivatives can be activated and functionalized with metal and metal nanoparticles, polymers, drugs, and other small molecules. They can be used in various research fields including regeneration, drug delivery, gene delivery, protein delivery, nanoparticles release, cell and tumor imaging, physiochemical properties development of other materials, multimodal bioimaging, and cancer therapy. Figure 1 demonstrates the diverse applications of graphene materials in caries and periodontal disease management.

### 2.2. Literature Survey

This review includes studies published in the English language on graphene and graphene derivatives and their functionalized composites used in caries and periodontal disease management. Abstracts, editorials, letters, and literature reviews were excluded. The EMBASE, Web of Science, Google Scholar, and PubMed databases were systematically searched. In the search, the keywords consisting of “graphene”, “caries,” “demineralization”, “remineralization”, “periodontal”, “regeneration”, “tissue engineering” and “bone regeneration” were employed. These keywords would cover information about graphene and its derivatives in caries and periodontal disease management. After reading all the manuscripts, the obtained articles were selected to describe in the terms of caries management and periodontal disease management.

## 3. Result and Discussion

### 3.1. Graphene Derivatives in Caries Management

Dental caries is a highly prevalent disease. Cariogenic biofilms are mainly responsible for dental caries. Caries initiates with the chemical dissolution of dental hard tissue by the acid produced through dietary carbohydrate metabolism of bacteria that adhered (as biofilm) to the tooth surface. A prolonged stagnation of biofilm enhances enamel and dentin desolation and progresses to cavity formation on the tooth surface [50,51]. These biofilms are the organized colony of microbial communities enclosed in an extracellular cohesive matrix (i.e., extracellular polysaccharide) where *Streptococcus mutans* is the main cariogenic pathogen. They produce insoluble extracellular polysaccharides, which facilitate bacterial growth and the formation of cariogenic biofilms. This is why most of the research focuses on developing biomaterials to inhibit *Streptococcus mutans* [52].

Remineralization of demineralized caries also can stop caries progression. It is said that maximum mineralization in the human body is seen in teeth by continuous demineralization and remineralization throughout life with varying amounts to maintain tooth integrity [53]. It breaks if demineralization suppresses the remineralization and results in caries progression [54]. Therefore, either stopping the biofilm formation, remineralization of demineralized hard tissue, or a combination of both is the scientifically logical point of view for caries prevention.

Although caries risk assessment and remineralization of initial lesions have controversy, diverse advanced research and nanotechnology have developed risk-specific biomaterials or board functional nano-biomaterials and opened the doors to caries prevention [55]. Graphene’s antibacterial effect became known first in 2010 and was widely explored afterward for various applications [20,56]. Currently, graphene has attracted much attention in caries research as a preventive, cariostatic, and remineralizing material. Research has well demonstrated that graphene derivatives are significant in inhibiting cariogenic bacteria, preventing dental hard tissue demineralization, and facilitating remineralization.

#### 3.1.1. Application against Cariogenic Pathogens

Although graphene and its derivatives can inhibit cariogenic bacteria, most of them are studied together with antimicrobial metals or non-metal or polymer nanoparticles, such as copper, silver, zinc, peptides, and polymer nanoparticles, to improve the antibacterial properties or facilitate the sustainable release of incorporated nanoparticles [57,58,59,60,61,62,63,64,65,66,67,68,69,70,71]. They are also studied with existing dental materials, especially incorporated into restorative cements, either to improve antibacterial properties against *Streptococcus mutans*, reduce dental hard tissue demineralization, or facilitate remineralization [60,63,64,66,72,73,74].

Graphene nanosheet was reported as very effective against *Streptococcus mutans* [63]. Simultaneously, graphene oxide nanosheets were demonstrated effective in reducing *Streptococcus mutans* [60,61]. Subsequently, metal functionalized graphene materials, graphene-silver nanoparticles, were also found effective against *Streptococcus mutans* without any significant cytotoxicity [13,57,64,65]. Similarly, reduced graphene-silver nanoparticles and graphene-nanoplatelets doped silver nanoparticles showed an antibiofilm effect against *Streptococcus mutans* biofilm [58]. Moreover, graphene oxide-copper nanocomposites reduced *Streptococcus mutans* growth significantly [62]. In addition, graphene-zinc nanocomposites were effective in reducing *Streptococcus mutans* biofilm. There are also reports of suppressing acid production and glucan formation, which are responsible for caries and biofilm formation [59].

In other studies, amino-functionalized graphene oxide was reported with potential against cariogenic bacteria *Streptococcus mutans* [67,70]. At the same time, graphene oxide-coated zirconia was also reported to inhibit *Streptococcus mutans* [69]. Some other studies reported that poly methyl methacrylate incorporated graphene oxide can greatly inhibit *Streptococcus mutans* growth [72,74]. Another reported that fluorinated graphene also can inhibit *Staphylococcus aureus* and *Streptococcus mutans* [73]. Some other studies reported that after treating with graphene oxide, graphene oxide-carnosine, and graphene oxide-carnosine-hydroxyapatite the survival rate of *Streptococcus mutans* was significantly reduced [13,75].

Although graphene oxide and antisense vicR could significantly inhibit biofilm and extracellular polysaccharide production alone, graphene oxide–polyethyleneimine–antisense vicR was reported as superior in inhibiting extracellular polysaccharide regulation, virulence-associated gene expression, and biofilm formation. Therefore, the study suggested that graphene oxide–polyethyleneimine–antisense vicR ribonucleic acid could be a highly potent agent for caries prevention [76]. On the other hand, one study reported that graphene oxide could be a potential nanocarrier. It was described that the functionalization of graphene oxide with antimicrobial photodynamic therapy can significantly enhance indocyanine green loading and stability, and could enhance the inhibitory effects against *Streptococcus mutans* [75]. Interestingly, peptide-functionalized reduced graphene oxide nanocomposite was also reported to inhibit cariogenic bacteria [68].

By overseeing all these studies, it can be hypothesized that either pristine or functional nanocomposites of graphene and its derivatives could potentially be used against cariogenic bacteria. However, the established mechanism of antibacterial activities of graphene derivatives is still to be explored. Several antibacterial mechanisms have been described to demonstrate graphene and its derivatives in inhibiting cariogenic microbe and their biofilm. Physical damage, membrane stress, oxidative stress, and electron transfer were well considered [77]. Therefore, advanced studies should be performed to translate the standard anticaries mechanism of graphene derivatives for caries management in clinical settings.

#### 3.1.2. Application for Tooth Remineralization

As a consequence of the *Streptococcus mutans* acidic by-product, demineralization initiates dental caries. At the same time, the counteraction of remineralization protects teeth from decay. Graphene can facilitate remineralization. In a study, graphene-fluorine was reported to enhance the remineralization of white spot lesions [78]. At the same time, graphene oxide fluorhydroxyapatite was also reported to prevent demineralization by resisting hydroxide dissolution [79].

In several studies, graphene oxide conjugated bioactive glass was reported to significantly increase the anti-demineralization effect, microhardness, shear bond strength, and adhesive remnant index with no or low cytotoxicity [13,80]. Graphene oxide and montmorillonite were reported to exhibit enhanced mechanical properties and bioactivity while incorporated in resin-based composite [81]. Interestingly, multi-walled carbon nanotube/graphene oxide hybrid carbon-based nanohydroxyapatite was reported to protect against dentin erosion [82].

In one study, reduced graphene oxide-silver was found to reduce enamel surface roughness and mineral loss, thus reducing the lesion depth [83]. Moreover, another study showed that graphene oxide could be a bioceramic support material to enhance hydroxyapatite deposition [84]. In addition, graphene oxide quantum dot incorporated mesoporous bioactive glass was reported to show excellent dentinal sealing and rapid mineralization. They promoted hydroxyapatite formation without interfering with calcium, silicon, and phosphate ions release [85]. Although there are potentials and limitations of graphene and its derivatives on antimicrobial effect, remineralization, and dual action there is no strong clinical evidence; therefore, advanced investigations are required to validate the optimal outcome and clinical applications.

Table 1 shows the investigated results and potential applications of graphene materials in caries management. The antimicrobial activity, remineralization, or dual action excels graphene and its derivatives to be potential candidates in advanced caries research. In the future, advanced translational research will be evidence to translate graphene materials into clinical applications.

### 3.2. Graphene Derivatives in Periodontal Disease Management

Periodontal diseases are highly prevalent among other dental diseases next to dental caries [91]. Periodontitis and peri-implantitis are chronic inflammatory conditions that infect the periodontium around the tooth or implant. They are also considered autoimmune oral diseases [92]. Although an immune response against periodontal pathogens is usually found in periodontitis or implantitis, the constant invasion of periodontal pathogens impairs the host’s innate and acquired immunity resulting in periodontal tissue destruction, including alveolar bone [93]. Therefore, controlling infection, preventing tissue destruction by eliminating pathogens and biofilms from the tooth or implant surface, and reducing tissue invasion are the main objectives of periodontitis and peri-implantitis treatment [94,95]. Table 2 shows the perspectives of graphene materials in the management of periodontal disease regarding their antimicrobial activity, periodontal tissue, and bone regeneration.

#### 3.2.1. Application against Periodontal Pathogens

*Porphyromonas gingivalis* is known as the primary pathogen for periodontal diseases. Most research is targeted to develop novel biomaterials or drugs to inhibit *Porphyromonas gingivalis* [96]. In recent decades, graphene materials were well investigated against periodontal pathogens. In one study, graphene-reinforced titanium was assessed against oral pathogens (*Streptococci mutans*, *Fusobacterium nucleatum*, and *Porphyromonas gingivalis*) and reported to have a high inhibitory effect on *Porphyromonas gingivalis* [97].

Another study investigated graphene silver polymethyl methacrylate on *Porphyromonas gingivalis* and *Enterococcus faecalis* and reported to have enhanced antibacterial effects [98]. At the same time, graphene-coated Ti-6Al-4V alloy was assessed on *Porphyromonas gingivalis*, *Fusobacterium nucleatum,* and *Candida albicans*. They applied an oxidative stress-induced antimicrobial mechanism by measuring the reactive oxygen species (ROS) generation and were reported to exhibit antimicrobial effects against these oral pathogens [99]. Another study showed that an ultrathin film of graphene oxide and lysozyme composite on a titanium surface exhibited antibacterial activities against *Staphylococcus aureus* and *Escherichia coli* [100].

Graphene oxide nanosheets were demonstrated as effective against *Porphyromonas gingivalis* and *Fusobacterium nucleatum.* The study showed that graphene oxide nanosheets could penetrate and destroy the cell wall and membrane, and initiate plasma leakage, resulting in cell death [60]. Zinc oxide functionalized graphene oxide polyetheretherketone was reported as an antibacterial agent against periodontal pathogens and biofilms [101,102]. Another study demonstrated that DNA-aptamer-nanographene oxide was highly effective in reducing *Porphyromonas gingivalis* [103].

Conversely, the use of graphene oxide followed by brushing was reported to be effective in eliminating *Streptococcus mutans*, *Fusobacterium nucleatum*, and *Porphyromonas gingivalis* and their biofilms [104]. In one study, graphene oxide wrapped under mineralized collagen for photothermal therapy, as its antimicrobial effect, was found effective against *Streptococcus sanguinis*, *Fusobacterium nucleatum*, and *Porphyromonas gingivalis* [105]. Moreover, minocycline hydrochloride-loaded graphene oxide was also reported to have antibacterial activities against *Staphylococcus aureus* and *Escherichia coli* [106]. On the other hand, graphene oxide nano-coated titanium was reported to exert a long-term persistent inhibitory effect on the *Candida albicans* biofilm [107].

Studies showed that carbon quantum dots can inhibit *Porphyromonas gingivalis* biofilm formation. Similarly, graphene oxide quantum dots and curcumin composite was also reported to inhibit polymicrobial biofilm formation, including periodontal pathogens such as *Aggregatibacter actinomycetemcomitans*, *Porphyromonas gingivalis*, *Prevotella intermedia,*
*Prevotella nigrescens, Escherichia coli,* and *Staphylococcus aureus* [108,109].

Observing all this research, it can be assumed that graphene and its derivatives could effectively prevent periodontal disease by inhibiting periodontal pathogens and their biofilms; however, further investigations are necessary to investigate these in vitro results in clinical settings.

#### 3.2.2. Application for Periodontal Tissue Regeneration

Despite the prevention or reduction of periodontal disease, the reproduction of lost periodontal tissue is highly expected, thus regenerative periodontal therapy is a developing research interest. Several techniques, including stem cells, scaffolds, biomaterials, or a combination were applied in various studies [110,111,112]. Generally, in regenerative dentistry, stem cells, growth factors, or their conjugates are usually delivered to the infected site using scaffolds or other nanocarriers. Among several other nano-biomaterials, graphene oxide-based materials, such as graphene oxide scaffolds [113,114], graphene oxide-gelatin [115], graphene oxide-fibroin [116], graphene oxide-collagen [117], graphene oxide-alginate [118], graphene oxide-chitosan [119], and graphene oxide-titanium substrates [120] were reported as promising in tissue engineering for their biocompatibility, bioactivity, enhance cell adhesion, and proliferation properties to periodontal ligament stem cells.

Graphene oxide-silk fibroin was reported as an excellent candidate for tissue engineering. In a study, a composite of graphene oxide-silk fibroin was investigated for cell adhesion, proliferation, viability, and mesenchymal phenotype expression of periodontal stem cells, and was reported to have high potential in the therapeutic capacity of this composite for regenerative dental applications [116]. On the other hand, an ultrathin film of graphene oxide and lysozyme composite on the titanium surface enhanced the osteogenic cell differentiation of human dental pulp stem cells [100].

In a study, minocycline hydrochloride conjugated graphene oxide was investigated in a peri-implantitis model and reported a very negligible bone loss in the minocycline hydrochloride-graphene oxide-titanium groups compared to that of titanium and minocycline hydrochloride-conjugated titanium group or the graphene oxide-titanium group. Moreover, there were almost no neutrophils found in graphene oxide-titanium and minocycline hydrochloride-graphene oxide-titanium groups, but a deposit of osteocyte cells was observed. Thus, suggested minocycline hydrochloride-conjugated graphene oxide could be a good therapeutic coating for preventing peri-implantitis [106].

Another study investigated the bioactivity of human periodontal ligament stem cells on graphene oxide-coated titanium substrate, which demonstrated: a higher cell proliferation rate and alkaline phosphatase activity; upregulated osteogenesis-related genes such as alkaline phosphatase, bone sialoprotein, collagen type I, and osteocalcin; a higher runt-related transcription factor 2 expression than that of on the control sodium titanium substrate [120]. Simultaneously, another study showed that the cell proliferation and osteogenic differentiation in the periodontal ligament stem cells were increased when they use poly (ε-caprolactone) conjugated graphene oxide scaffolds [121].

In addition, one study showed that a graphene oxide scaffold was biocompatible and capable to form new bone and periodontal tissue [113]. In a separate study, it was suggested that both the physical and chemical properties of graphene play a role in cell differentiation. In their study, periodontal ligament stem cells were seeded on graphene scaffolds and they observed that bone-related genes and proteins (collagen I, runt-related transcription factor 2, and osteocalcin) were upregulated on graphene scaffolds [122].

At the same time, reduced graphene oxide has also shown potential for stem cell proliferation and is suggested for use in regenerative dentistry [123,124]. One study showed that reduced graphene oxide-incorporated chitosan nanocomposites were compatible with the adhesion, proliferation, and osteogenic and neurogenic cell differentiation of human mesenchymal stem cells [123]. Another study showed that reduced graphene oxide exhibited high stimulation to the osteogenic proliferation, differentiation, and mineralization of human mesenchymal stem cells [124]. In another study, silk-fibroin was incorporated into graphene oxide/reduced graphene oxide to develop a bilayer composite. These bilayer biocomposites were found to enhance cell proliferation and the osteoblastic and cementoblastic cell differentiation of periodontal ligament stem cells. In addition, the alkaline phosphatase, the osterix, and runt-related transcription factor 2 levels were also found to increase with overexpression of cementum protein I [125].

Graphene oxide and reduced graphene oxide have been included in a porous titanate scaffold on titanium implants to develop a drug delivery system for dexamethasone, the study of which showed that both dexamethasone-graphene oxide-titanium and dexamethasone-reduced graphene oxide-titanium increases alkaline phosphate activity in the bone marrow mesenchymal stem cells. They exhibit osteogenic genes, osteopontin and osteocalcin expression, and enhanced osteogenic activity [126]. Moreover, reduced graphene oxide-coated hydroxyapatite was reported to exhibit murine preosteoblastic osteogenic cells’ differentiation to bone lineage without any interference from osteogenic differentiation factors.

Reduced graphene oxide-coated hydroxyapatite also showed an osteogenic differentiation of human mesenchymal stem cells. Therefore, they suggested applying this composite for orthopedic and dental regeneration [127]. Graphene oxide quantum dots were also investigated on human periodontal ligament stem cells. In a fluorescent labeling application study, the graphene oxide quantum dots showed that they could enter the cell membrane and increase fluorescence intensity without any toxicity. This was suggested as a potential for live cell labeling of human periodontal ligament stem cells [128].

There have been tremendous advancements in tissue regenerative research using graphene derivatives; however, to date, no clinical trial is reported. More in vitro, in vivo, animal trials, and ultimately a human trial should be performed to validate the optimal result for applying these novel materials in clinical settings.

#### 3.2.3. Application for Periodontal Bone Regeneration

Periodontal bone loss usually occurs due to several factors, including tooth extraction, trauma, infection, systemic or local alterations in the host response, malignancy, or multifactorial causes [129]. Bone substitutes are widely used to replace bone loss including bone grafts (autograft, allograft, and xenograft), ceramics (hydroxyapatite, tricalcium phosphate, and calcium sulfate), and growth factors (demineralized bone matrix, platelet-rich plasma, and bone morphogenic proteins) [130]. Simultaneously, implants are commonly used for replacing lost teeth. Implant success mainly depends on osseointegration and bone regeneration. Osseointegration is the “benchmark” for assessing implants’ success, and the interactions between the implants and host cells [131]. Therefore, the surface modification of implant materials is well considered for property development. The graphene derivative graphene oxide and its functional conjugates exhibited promising results to improve bioactivity and osseointegration [132,133].

Studies have well reported that graphene can promote the osteogenic differentiation of several cells, including osteogenic cells and stem cells (bone marrow stem cells, periodontal ligament stem cells, and dental pulp stem cells), and have suggested applying this in different forms (composite, scaffolds, and coatings) [134]. A study using graphene coating on titanium demonstrated that graphene-titanium can inhibit multiple oral pathogens and enhance human gingival fibroblast growth [97]. Another study found that single-layer graphene sheets on titanium exhibited excellent adhesion, proliferation, and osteogenic differentiation of human gingival fibroblasts, human adipose-derived stem cells, and human bone marrow mesenchymal stem cells. In addition, it exhibited enhanced antibacterial properties [135]. Moreover, graphene was reported to stimulate different types of stem cells to differentiate into osteoblasts. For example, the graphene-hydroxyapatite composite sheet showed good biomimetic mineralization and osteogenic differentiation [136,137]. Table 2 summarizes graphene and its derivatives for the management of periodontal disease

The chitosan-incorporated graphene oxide-chitosan-hydroxyapatite coating on titanium substrates was reported to show improved in vitro and in vivo osseointegration [138]. Moreover, some other studies used (3-Aminopropyl) triethoxysilane-induced graphene coatings to enhance cell attachment, proliferation, and osteogenic differentiation [120,139,140]. At the same time, graphene oxide can enhance bone marrow stem cells’ adhesion, proliferation, osteogenic differentiation, and bone-implant interaction. New bone formation and the proximity between implants and bone tissue were observed in a study using graphene oxide coating on the implant surface [141]. In one study, graphene oxide-coated titanium showed capabilities in enhancing dental pulp stem cell differentiation into the osteoblastic cell lineage and gene expression for osteoblasts [142]. Moreover, graphene oxide was reported as a promising carrier for a drug delivery system, for example, delivering therapeutic proteins (bone morphogenetic protein-2 and bone sialoprotein) [143,144].

Graphene oxide-coated titanium substrates, loaded with bone morphogenetic protein-2, exhibited osteogenic differentiation of human bone marrow-derived mesenchymal stem cells, new bone formation, and sustained drug delivery without interfering with drug structure and bioactivity [143]. In addition, in another study, the researchers showed that graphene oxide is a good carrier for the delivery of bone morphogenetic protein-2 and substance P. They demonstrated that this conjugated delivery enhanced new bone formation. In addition, it was also reported that bone morphogenetic protein-2 and substance P delivery enhanced alkaline phosphatase activity and osteoblastic activity, which can increase osteointegration [144]. On the other hand, a new injectable graphene oxide hydrogel with chitosan was reported as a mechanically strong stem cell scaffold for bone regeneration [145].

Similarly, reduced graphene oxide also has the potential as a surface coating on titanium implants. To show the osteogenic differentiation of reduced graphene oxide, a study used collagen type I conjugated reduced graphene oxide and found collagen–graphene oxide/reduced graphene oxide can activate the differentiation of human bone marrow stem cells into osteoblasts [146]. Comparably, new bone regeneration using reduced graphene oxide was well reported in one study [127].

Another study showed that multipass caliber rolled titanium alloy coated with dexamethasone-loaded reduced graphene oxide enhanced osteogenic cell growth and osteoblastic differentiation of MC3T3-E1 cells. In addition, a reduced graphene oxide-coated titanium implant was implanted in an artificial bone block and reported to be stable [139]. On the other hand, sandblasted/acid-etched titanium implants coated with reduced graphene oxide and bone morphogenetic protein-2 were reported to exhibit enhanced cell proliferation, increased alkaline phosphatase activity, promoted matrix mineralization, improved expression of osteogenesis-related genes and protein, and improved osseointegration; thus, this is suggested as a promising implant material [147].

In a study, reduced graphene oxide and pulsed electromagnetic fields were combinedly used to evaluate osteogenesis and the neurogenesis of mechanical stem cells. The investigation showed that the conjugates enhanced osteogenic, neurogenic, and adipogenic differentiation of human alveolar bone marrow stem cells. They suggested using this conjugate for stem cell and tissue engineering [148].

**Table 2 pharmaceutics-14-01997-t002:** Graphene and its derivatives for the management of periodontal disease.

*Graphene and Its Derivatives*	*Properties [Ref* *erence(s)* *]*
** *Graphene* **	
*Graphene*	Inhibits oral fungi biofilm [107] Inhibits oral periodontal pathogenic bacteria [98] Increases bone regeneration [122,135,137,140]
*Graphene-hydroxyapatite*	Increases bone regeneration [136]
*Graphene-titanium*	Inhibits periodontal pathogenic bacteria [97,99]
** *Graphene Oxide* **	
*Graphene Oxide*	Inhibits periodontal pathogenic biofilms [60,103,104,105] Increases bone regeneration [104]
	Increases periodontal tissue regeneration [113,121]
*Graphene Oxide* *-* *p* *olyetheretherketone*	Increases bone regeneration [145]
*Graphene Oxide-chitosan*	Inhibits periodontal pathogenic biofilms [101,102]
*Graphene Oxide-hydroxyapatite*	Increases bone regeneration [132]
*Graphene Oxide-silk fibroin*	Increases bone regeneration [116,125]
*Graphene Oxide-titanium*	Increases bone regeneration [120]
*Graphene Oxide-chitosan-hydroxyapatite*	Increases bone regeneration [138]
*Graphene Oxide-lysozyme-titanium*	Inhibits oral bacteria [100] Increases bone regeneration [100]
*Graphene Oxide-minocycline hydrochloride-titanium*	Inhibits oral bacteria [106] Increases bone regeneration [106]
*Graphene Oxide-dexamethasone-titanium*	Increases bone regeneration [126]
*Graphene Oxide-bone morphogenetic protein 2-titanium*	Increases bone regeneration [143]
** *Reduced Graphene Oxide* **	
*Reduced Graphene Oxide*	Increases bone regeneration [146,148]
*Reduced Graphene Oxide-chitosan*	Increases bone regeneration [123]
*Reduced Graphene Oxide-hydroxyapatite*	Increases bone regeneration [127]
*Reduced Graphene Oxide-titanium*	Increases bone regeneration [147]
*Reduced Graphene Oxide-dexamethasone-titanium*	Increases bone regeneration [126]
** *Graphene Oxide* ** ** *Q* ** ** *uantum Dots* **	
*Graphene Oxide Quantum Dots*	Living cell labeling [128]
*Graphene Oxide Quantum Dots-curcumin*	Inhibits periodontal pathogenic bacteria [109]

## 4. Conclusions

In conclusion, in vitro studies demonstrated that graphene and its derivatives have antimicrobial and remineralizing properties for the management of dental caries and periodontal disease. Therefore, advanced research is essential to translate their application to clinical dentistry.

## Figures and Tables

**Figure 1 pharmaceutics-14-01997-f001:**
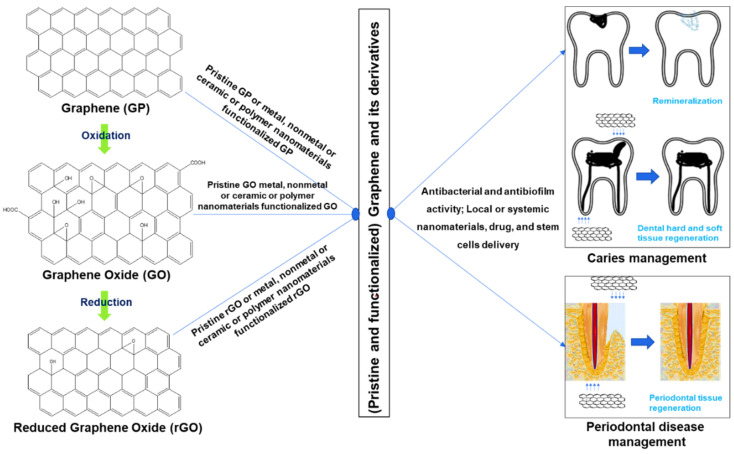
Use of graphene for management of dental caries and periodontal disease.

**Table 1 pharmaceutics-14-01997-t001:** The properties of graphene and its derivatives for the management of dental caries.

*Graphene and Its Derivatives*	*Properties [Ref* *erence(s)* *]*
** *Graphene* **	
*Graphene*	Inhibits cariogenic biofilm [63,66]
*Graphene-silver nanoparticles*	Inhibits cariogenic biofilm [58]
*Graphene-zinc nanoparticles*	Inhibits cariogenic biofilm [59]
*Graphene-zinc oxide nanoparticles*	Inhibits cariogenic biofilm [86]
*Graphene-fluorine*	Inhibits cariogenic biofilm [73,78] Promotes enamel and dentin mineralization [78]
** *Graphene Oxide* **	
*Graphene oxide*	Inhibits cariogenic bacteria [13,60,61,67,69,70] and fungi [87] Inhibits cariogenic biofilm [69,76,88,89] Promotes enamel and dentin mineralization [79,81,82,84]
*Graphene oxide-silver nanoparticles*	Inhibits cariogenic bacteria [13,57,64,65]
*Graphene oxide-bioactive glass*	Inhibits cariogenic bacteria [80] Promotes enamel and dentin mineralization [80]
*Graphene oxide-silver-calcium fluoride*	Inhibits cariogenic bacteria [13]
*Graphene oxide-carnosine-hydroxyapatite*	Inhibits cariogenic bacteria [75]
*Graphene oxide-copper*	Inhibits cariogenic biofilm [62]
*Graphene oxide-polyethyleneimine*	Promotes enamel and dentin mineralization [90]
*Graphene oxide-poly-methyl methacrylate*	Inhibits cariogenic bacteria [72,74]
*Graphene oxide-nanoribbon*	Inhibits cariogenic biofilm [71]
** *Reduced Graphene Oxide* **	
*Reduced graphene oxide*	Inhibits cariogenic bacteria [68]
*Reduced graphene oxide-silver nanoparticles*	Inhibits cariogenic biofilm [64] Promotes enamel and dentin mineralization [83]
** *Graphene Oxide Quantum Dots* **	
*Graphene oxide quantum dots-bioactive glass*	Promotes enamel and dentin mineralization [85]

## Data Availability

Not applicable.

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
