# Peer review of "In Vitro Studies of Graphene for Management of Dental Caries and Periodontal Disease: A Concise Review"

_pharmaceutics, 2022, doi:10.3390/pharmaceutics14101997_

Round 1
Reviewer 1 Report
There are some clarifications to be made:
Line 18-19- it's hard to understand, maybe it's better: ... "in caries prevention, because of their antibacterial properties, and in regenerative dentistry".
Line 20- Eliminate "and Candida albicans" (because it appears to be referred to as a cariogenic microbe) or better explain the roles of this yeast.
Author Response
General Comments: There are some clarifications to be made:
Our response: Thank you so much. We appreciate your comments. We have revised it accordingly.
Comments 1: Line 18-19- it's hard to understand, maybe it's better: ... "in caries prevention, because of their antibacterial properties, and in regenerative dentistry".
Response 1: Thank you for your comment. We have revised the abstract. Revised parts of the abstract are highlighted in green in the main manuscripts.
Comments 2 Line 20- Eliminate "and Candida albicans" (because it appears to be referred to as a cariogenic microbe) or better explain the roles of this yeast.
Response 2: Thank you for your suggestion. We have eliminated "and Candida albicans" from the abstract.
Reviewer 2 Report
Thank you for submitting The use of graphene for management of dental caries and peri- 2 odontal disease: A concise review. The aim of this paper is to provide a concise review of the potential application of graphene materials in caries and periodontal disease management. As graphene is a new material that is increasingly being used, it is an important review to do.
The manuscript’s results are not reproducible based on the details given in the methods section. There is not a method section. Even if it is a review, it is necessary to specify how the search was conducted, on what date, what keywords were used, inclusion and exclusion criteria. You should restructure all the manuscript sections, where the introduction finish, add material and method, add results and finally the discussion and conclusions section. You should add a diagram flow of the selection of the studies. I will check the article again when it is well structured.
Without references, the % of similarity is 21%, it should be less than 20% for being acceptable.
If you only selected laboratory studies, you should specify this on the title.
Which are the limitations of your study?
Author Response
Comments 1: The manuscript’s results are not reproducible based on the details given in the methods section. There is not a method section. Even if it is a review, it is necessary to specify how the search was conducted, on what date, what keywords were used, inclusion and exclusion criteria. You should restructure all the manuscript sections, where the introduction finish, add material and method, add results and finally the discussion and conclusions section. You should add a diagram flow of the selection of the studies. I will check the article again when it is well structured.
Response 1: Thank you for your comment. We have added the “Materials and method” and “Result and Discussion” sections in the manuscripts for better understanding. We have added “literature survey” in the main manuscripts. It’s a concise review, which is why we didn’t include a flow diagram.
Hope it is better now. We have rearranged the manuscripts and structural changes are highlighted in yellow.
Comments 2: Without references, the % of similarity is 21%, it should be less than 20% for being acceptable.
Response 2: Thank you for your suggestion. We have revised the entire manuscript thoroughly to avoid unwanted redundancy and similarity issues.
Comments 3: If you only selected laboratory studies, you should specify this on the title.
Response 3: Thank you for your recommendation. We have revised the title accordingly. The present title is “ In vitro studies of graphene for management of dental caries and periodontal disease: A concise review”
Comments 4: Which are the limitations of your study?
Response 4: Thank you for your incisive comment, this review is based on in vitro
investigations. At this stage, we can't give any definite conclusion of the results that can be translated into the clinical settings.
Round 2
Reviewer 2 Report
The instructions given by the reviewer have not been followed.
What day was the search conducted?
How many articles were reviewed? How many were read in full text?
I asked you to add a flowchart, even if it is a concise review, I want to know how many articles were analysed to make it.
How were the words put together for the search? How much came up in each search?
Section 2.1 is not a material and methods section in a review.